

# Evaluation of a biosecurity survey approach for contamination by *Chlamydia pecorum* in koala rehabilitation, field capture, and captive settings

Andrea Casteriano[1], Astrid Robin Van Aggelen[2], Shali Fischer[2], Cheyne Flanagan[2], Caroline Marschner[1], Hannah Nugent[3], Wilhelmina Huston[4] and Damien P. Higgins[1]

[1] Faculty of Science/Sydney School of Veterinary Science, University of Sydney, Sydney, New South Wales, Australia
[2] Koala Hospital Port Macquarie, Koala Conservation Australia, Port Macquarie, New South Wales, Australia
[3] Faculty of Science/School of Life Sciences, University of Technology Sydney, Sydney, New South Wales, Australia
[4] Faculty of Science, University of Technology Sydney, Sydney, New South Wales, Australia

Corresponding author
Andrea Casteriano,
andrea.casteriano@sydney.edu.au

## ABSTRACT

Transmission of *Chlamydia pecorum* between koalas is a potential risk in field capture or rehabilitation settings, where koalas are held in proximity to each other, or equipment is shared between animals. Given the impact of *C. pecorum* on koala welfare and population viability it is surprising that quarantine and disinfection protocols in a koala rehabilitation facility or capture settings have not previously been evaluated. This study aimed to evaluate an approach, based on the detection of chlamydial DNA and cell viability, to determine the degree of environmental contamination within a koala care facility. Various fomite sites associated with koala care at a koala rehabilitation facility in New South Wales, Australia were identified as potential sources of chlamydial contamination, following exposure to koalas known to be infected with *C. pecorum*. Fomite sites were swabbed following exposure, and again after decontamination procedures were carried out. Samples were tested for the presence of chlamydial DNA using qPCR and viability using both RT-qPCR and cell culture. From a total of 239 sampling events, 30 tested qPCR positive for chlamydial DNA, with 19 and 11 samples corresponding to pre-decontamination and post-decontamination events respectively. Detection of chlamydial DNA appeared to be most common in the examination room, especially on fomite sites in direct contact with koalas. Physical removal of chlamydial DNA, or its degradation by the elements, appeared to be more common on outdoor enclosures, clothing, and hands. Based on the cell culture assay, of the pre-decontamination samples with chlamydial DNA, eight had viable chlamydial cells, two of these at low levels. Of the post-decontamination samples with chlamydial DNA, one had a moderate number, and one had a very low number of viable chlamydial cells. RT-qPCR was unsuccessful in determining cell viability due to low yields of RNA and high levels of contaminants from the environmental samples. The outcomes of this study provide a knowledge base for the design of future biosecurity evaluation guidelines in captive and koala rehabilitation facilities. The higher incidence of chlamydial DNA detection

by qPCR than viable organism highlights the need to use viability assays in similar studies. However, further investment is still needed to optimise these methods and improve sensitivity for complex environmental samples.

## INTRODUCTION

As is the case for many wildlife diseases, potential for transmission of *C. pecorum* between koalas in care poses a risk to koala welfare, and to wild populations. Hundreds of koalas are admitted into care facilities every year, with chlamydiosis being the most common reason for hospital admission in Northern New South Wales (*Lunney et al., 2022*) and Southeast Queensland (*Gonzalez-Astudillo et al., 2017*), and a major cause of infertility and population decline (*Fabijan et al., 2019*). Chlamydial strain diversity exists across the fragmented catchments of koala hospitals (*Higgins et al., 2012*) and between neighbouring populations (*Fernandez et al., 2019*; *Robbins et al., 2020*) and, in some areas, *Chlamydia*-free populations exist that require protection (*Fabijan et al., 2019*). Certain *C. pecorum* genotypes have been associated with disease progression, while others have been associated with the absence of clinical disease (*Robbins et al., 2020*). Therefore, effective biosecurity practices are necessary to prevent transmission of *C. pecorum* between research populations, or across hospital catchments following fomite transmission between koalas in care. Environmental contamination, and disinfection protocols, have not been evaluated or standardised in koala care facilities. The potential for fomite transmission of *Chlamydia trachomatis* (*Versteeg et al., 2020*), *Chlamydia suis* (*De Puysseleyr et al., 2014*) and *Chlamydia pneumoniae* (*Falsey & Walsh, 1993*) has been previously evaluated using PCR and cell cultures to detect *Chlamydia* and its viability on various fomite surfaces. Although self-inoculation or direct contact of mucosal membranes with the contaminated surface is still required for transmission (*Gitsels, Sanders & Vanrompay, 2019*), *C. pneumoniae* and *C. trachomatis*, at least, can maintain viability in the environment for 24–30 h (*Falsey & Walsh, 1993*; *Versteeg et al., 2020*). Guidelines for disinfection of *Chlamydia psittaci* exist for avian enclosures and equipment (*Balsamo et al., 2017*), but these have not been evaluated for *C. pecorum* or for the equipment commonly used in koala care facilities, such as soiled towels and natural bark-covered timber perches. Avoiding pathogen transmission by using various methods of decontamination and segregation is a large part of the daily work in koala care facilities, though current disinfection processes are not evaluated, and clinicians cannot be assured if risks are being mitigated.

Due to the novel fomite sites encountered in koala care facilities, this study aimed to evaluate an approach, based on qPCR, cell culture and RNA-based viability assays, to determine the degree of environmental contamination by *C. pecorum* within a koala care facility. This study provides an evidence base for effective design of programs to monitor contamination and evaluate efficacy of biosecurity practices for *C. pecorum* in koala care

facilities, as well as preliminary evaluation of existing practices used in one best-practice facility.

# MATERIALS AND METHODS

## Sample collection

A total of 239 swabs in duplicate were collected at a koala rehabilitation facility in New South Wales, Australia, from various fomite sites associated with koala care (Table 1), using one DNA/RNA Shield SafeCollect™ Swab Collection Kit (Zymo Research cat# R1160-E) for both DNA detection and RT-qPCR, and one UTM® Transport Media swab (COPAN cat# 359C) for *Chlamydia* cell culture. Samples were collected before and after decontamination procedures (Table 1), from fomite sites identified in consultation with staff as potential sources of chlamydial contamination in the rehabilitation facility, following immediate exposure to koalas verified to be shedding *C. pecorum* based on point-of-care testing (*Hulse et al., 2019*). DNA/RNA Shield SafeCollect™ swabs were stored at −20 °C, while UTM® Transport Media swabs were stored in liquid nitrogen until processing.

Human volunteers for sampling of hands and clothing provided signed written consent prior to samples being collected. This study was approved by the Human Research Ethics Committee, University of Sydney (Approval No. 2022/007).

## Detection of chlamydial DNA

Genomic DNA from DNA/RNA Shield SafeCollect™ swabs was extracted using a MagMAX CORE Nucleic Acid Purification Kit (Thermo Fisher cat# A32702; Thermo Fisher Scientific, Waltham, MA, USA) by taking 200 μL of collection solution from each tube and adding it to a 1.5 mL tube containing 350 μL of MagMAX CORE Lysis Solution and 10 μL of Proteinase K, then briefly vortexed and incubated at 56 °C for 1 h. The lysate was then added to a 96DW-plate containing 350 μL of MagMAX CORE Binding Solution and 20 μL of MagMAX CORE Magnetic Beads, then immediately processed on a KingFisher™ Flex automated extraction instrument, using the MagMax_Core_Flex protocol. DNA was eluted to a final volume of 100 μL. Real-time PCR used SensiFAST™ Probe No-ROX Kit (Bioline cat# BIO-86005) on a CFX96 Touch™ Real-Time PCR Detection System with the corresponding CFX Maestro software (BioRad, South Granville, NSW, Australia) and 2 μL of template DNA. The qPCR reaction at a final volume of 20 μL included primers and probes targeting *C. pecorum* (*ompB* gene), *Chlamydia* genus (23S rRNA) and koala β-actin reference gene. All primers and probes, described elsewhere (*Hulse et al., 2018*), were used at a final concentration of 400 nM and 200 nM respectively. PCR conditions were 3 min at 95 °C, followed by 40 cycles of 10 s at 95 °C and 40 s at 58 °C. The limit of detection (LOD) of this multiplex assay is 86 copies of *C. pecorum* target per reaction. Samples for which *C. pecorum* was detected by qPCR were processed for cell viability RT-qPCR and their corresponding UTM® Transport Media swabs were processed for chlamydial cell culture.

**Table 1 Sites and associated decontamination procedures in the study.**

| Site sampled | Decontamination procedure |
|---|---|
| **Examination room** | |
| Bench | Spray and wipe down with F10® |
| Ultrasound probe | Wipe down with methylated spirits and paper towel |
| Stethoscope | Spray and wipe down with F10® |
| Oxygen mask | Washed with chlorhexidine then rinsed under running tap and dried with paper towel |
| Recovery basket | Sprayed with F10®, and wiped and air dried |
| Capture bag | Washed with laundry detergent and tumble dried |
| **Personnel** | |
| Clinicians' shoes | Foot bath (F10®) on entry and exit from clinic and enclosures |
| Clinicians' shirt | Washed with laundry detergent |
| Clinicians' hands | Washed with soap and water |
| Volunteer shoes | Foot bath (F10®) on exit from enclosure |
| Volunteer hands | Hand gel sanitisers (75% ethanol) on exit from enclosure |
| **Enclosures** | |
| Door handle | Not decontaminated during cleaning of enclosure |
| Walls | Hosed down with water |
| Mesh | Hosed down with water |
| Floor | Swept and hosed down |
| PVC browse holder | Hosed down |
| Towel under koala with urogenital disease (bleach wash) | Washed with laundry detergent plus a cup of bleach, tumble dried (standard procedure) |
| Towel under koala with urogenital disease (regular wash) | Washed with laundry detergent and tumble dried |
| Timber perches | Sprayed with F10®, hosed down and left outside in the elements |

**Note:**
F10® SC Veterinary Disinfectant (active ingredients quaternary ammonium compounds 5% and biguanide 8%). Soap = KLEENEX® Hand Cleansers. Laundry detergent = Easy Wash-Castle Chemicals PTY LTD. Hand gel sanitiser = Visbella® hand sanitiser (75% alcohol).

## Cell viability analyses

UTM® Transport Media swabs corresponding to chlamydial DNA-positive sampling events were inoculated onto McCoy B cells (ATCC cat #CRL-1696) in an established chlamydial culture system (*Lawrence et al., 2016*). Non-infected cell cultures and cells infected with *C. pecorum* at 1 MOI (1 IFU/cell) were included for use as negative and positive controls, respectively. At 44 h post-infection, a time point previously shown to be appropriate for the developmental cycle of *C. pecorum* (*Lawrence et al., 2016*), cells were fixed with methanol and stained to identify chlamydial inclusions (using FITC-conjugated anti-chlamydial LPS antibody) and viable cells *via* staining of the nuclei (DAPI). Cells were then imaged on an InCell imager and analysed for *C. pecorum* inclusions using ImageJ software (National Institutes of Health).

Samples collected using DNA/RNA Shield SafeCollect™ Swab Collection Kit that tested qPCR positive for *Chlamydia* were assessed in a RT-qPCR to further indicate viability. RNA was extracted from DNA/RNA Shield SafeCollect™ media for those samples using the QIAGEN Allprep Micro DNA/RNA extraction kit (cat #80284) as per the manufacturer's instructions. RNA samples were checked for concentration and purity on a

NanoDrop™ prior to cDNA synthesis. cDNA was synthesised from 5 µg of total RNA using the Invitrogen Superscript™ III First-Strand Synthesis System (cat # 18080051) as per the manufacturers guidelines. qPCR reactions were then prepared using 20 ng of cDNA, 200 nM of forward and reverse primers (Marsh et al., 2011) targeting the *ompA* gene of *C. pecorum* and Applied Biosystems Fast SYBR™ Green Master Mix (cat #4385610). DNA targets for normalising controls were included from the in-house assay. The reactions were then run on a Rotor-Gene Q system (QIAGEN). Results were analysed in the Rotor-Gene Q series software using endpoint analysis to compare Cq values of positive and negative controls. The presence of chlamydial RNA in this analysis signifies the potential for active, viable chlamydial cells.

## RESULTS

### Detection of chlamydial DNA

From the 239 sampling events, chlamydial DNA was detected in 30 samples (12.5%) using qPCR (Table 2). All positive results showed amplification for both the *Chlamydia* genus target and the *C. pecorum* specific target, and except for two samples (one hands and one oxygen mask), amplification was at or below the assay's LOD. Unexpectedly, no chlamydial DNA was detected on towels used under koalas with urogenital chlamydial disease, before or after decontamination. Among the samples collected, detection of chlamydial DNA was most common in the examination room (18/82, 21.9%), especially on fomite sites in direct contact with the koala, such as the oxygen mask, but also occurred on clothing and hands from personnel (7/56, 12.5%). All positive samples from enclosures (5/101, 4.9%) were from one indoor enclosure containing an infected koala, and one from its adjacent, also indoor, enclosure.

### Cell viability analyses

Based on cell culture, of the 19 pre-decontamination samples with chlamydial DNA, eight had viable chlamydial cells, two of these at low levels. Of 11 post-decontamination samples with chlamydial DNA, one had moderate levels (a volunteer's hands), and one had a very low number (a sample from an oxygen mask), of viable chlamydial cells (Table 2). It was not clear whether the latter sample was from a surface likely to be most exposed to cleaning and future patients, or a less exposed crevice.

The nature of the environmental samples had a detrimental effect on sensitivity of both viability assays. Some inhibition of *C. pecorum* inclusions/growth was detected, in association with contamination of samples by an array of other microbial species, including fungi, and bacteria. Two of the most contaminated samples were spiked with *C. pecorum* and when compared to a positive control inoculated with the same amount of *C. pecorum*, inhibition was observed in those samples (data not shown). Further, analyses of viability using the RT-qPCR method were unsuccessful due to low yields of RNA and high levels of contaminants from the environmental samples.

**Table 2 Number of chlamydial DNA positive (qPCR) and cell culture swabs before and after decontamination procedures.**

| Site sampled | Sampling events | DNA positive (qPCR) Before | After | Cell culture Before | After |
|---|---|---|---|---|---|
| **Examination room** | | | | | |
| Examination bench | 16 | 1 | 1 | – | – |
| Ultrasound probe | 16 | 1 | 2 | + | – |
| Stethoscope | 18 | 2 | 1 | – | – |
| Oxygen mask | 18 | 4 | 5[b] | +/++/–/– | +/–/–/–/NT |
| Recovery basket | 10 | 0 | 0 | NT | NT |
| Capture bag | 4 | 1 | 0 | +++ | NT |
| **Personnel** | | | | | |
| Clinicians' shoes | 8 | 0 | 0 | NT | NT |
| Clinicians' shirt | 8 | 3 | 0 | ++ | NT |
| Clinicians' hands | 4 | 1 | 0 | – | NT |
| Volunteer shoes | 14 | 0 | 0 | NT | NT |
| Volunteer hands | 22 | 2 | 1 | – | ++ |
| **Enclosures** | | | | | |
| Door handle | 8 | 0 | 0 | NT | NT |
| Walls | 30 | 2[a] | 0 | +++/++ | NT |
| Mesh | 8 | 0 | 0 | NT | NT |
| Floor | 10 | 1 | 0 | – | NT |
| PVC browse holder | 10 | 1 | 0 | +++ | NT |
| Towel under koala with urogenital disease (bleach wash) | 8 | 0 | 0 | NT | NT |
| Towel under koala with urogenital disease (regular wash) | 8 | 0 | 0 | NT | NT |
| Timber perches | 19 | 0 | 1 | NT | – |

Note:
NT, not tested; +, very few viable chlamydial organisms detected; ++, moderate numbers of viable chlamydial organisms detected; +++, many viable chlamydial organisms detected; –, no viable chlamydial organisms detected. [a]one sample in enclosure of infected koala, one adjacent. [b]no "before" sample available for one positive "after" sample.

## DISCUSSION

The study provided information of value in the design of guidelines for monitoring contamination and efficacy of disinfection practices for *C. pecorum* in koala care facilities and field studies. Common sites of contamination were identified, and decontamination practices appeared mostly effective in the limited samples available. Based on the detection of chlamydial DNA in absence of viable organisms, viability assays are clearly needed but our experience indicates further work is needed to find an optimal molecular viability assay for these complex environmental samples.

As expected, contamination was highest in sites associated with direct koala contact (oxygen masks and other clinical equipment) and in sites most protected from the elements (indoor enclosures). Future studies should prioritise these areas. While enclosures may be considered low risk, as chlamydial transmission is generally considered to require moist transfer to a mucosal surface (*Gitsels, Sanders & Vanrompay, 2019*), these conditions could be met by aerosolization of *C. pecorum* during hosing, especially within confined spaces protected from the elements. Oxygen masks may be considered higher risk due to potential conjunctival and respiratory contact. It was unclear whether positive
swabs were from the contact face of the mask or from crevices more hidden from disinfection and patient contact. Factors such as spray and wipe *versus* soaking, as well as disinfectant penetration and contact times, should be examined in future studies to confirm most effective approaches.

Decontamination practices appeared to significantly reduce risk by physical removal (reduction in DNA contamination) and disinfection (reduction in viability), though this was not absolute and viable *C. pecorum* was isolated from an anaesthetic mask in one instance despite thorough washing. Where crevices exist in clinical equipment such as masks, soaking rather than washing should be used, and disinfectant contact times strictly adhered to. Cleaning and disinfection are only one element of a biosecurity program and results of the study support the need for continuing segregation of animals, staff, and equipment-measures that are also employed at the facility studied.

The low prevalence and low chlamydial loads encountered in this study indicate that evaluation of disinfection and cleaning methods is likely best served by manipulative experiments, where surfaces are manually contaminated with known quantities of viable *C. pecorum*, rather than clinical sampling. Previous studies using *C. trachomatis* cultured stocks applied to various surfaces have demonstrated this to be a useful approach for evaluating not only sampling techniques but also the persistence of viable *Chlamydia* over time in the environment (*Novak et al., 1995*; *Versteeg et al., 2020*). More controlled manipulative experiments might also allow more effective evaluation of chlamydial viability, as both assays used for this purpose were adversely impacted by environmental contaminants. Clinical sampling would, however, remain useful to evaluate compliance and efficacy in local contexts once ideal methods are established.

Absence of detection of chlamydial DNA on towels that had been under animals with chlamydial urogenital disease and confirmed to be shedding *C. pecorum* at this anatomical site indicates that our approach to sampling this important fomite site may have not been effective. This finding is consistent with that of *Versteeg et al. (2020)* from a woven mat spiked with a *C. trachomatis* culture, suggesting the DNA or organisms may become bound to the substrate. In the current study, the towels were heavily soaked with urine, so it is possible that chlamydial organisms were carried through the towel by the urine, leaving insufficient DNA to allow recovery by swabbing the surface. Swabbing was used to replicate the type of contact expected between the material and the animal but excision of material for direct extraction may be more effective when evaluating efficacy of washing towels and bags.

Our results highlight the need for further optimisation or novel method development to evaluate chlamydial viability, due to the inhibition encountered with the environmental sample types associated with a koala care facility. Although qPCR is a sensitive screening tool, it does not discriminate between viable and non-viable *Chlamydia*. For example, an oxygen mask sample collected after decontamination had high qPCR chlamydial load, but no viable chlamydial cells were recovered (likely indicating successful disinfection but not removal of residual DNA). The uncertainty around viability assay sensitivity may have resulted in an under-representation of viable contamination in this study.

## CONCLUSIONS

The outcomes of this study provide a knowledge base for the design of future evaluation of biosecurity guidelines in captive and rehabilitation facilities; it is inappropriate to interpret this study as an assessment of transmission risk in the facility studied. The study supports that cleaning and disinfection are only one element of an effective biosecurity program; biosecurity practices at the facility studied go beyond these to include practices that ensure segregation of animals, staff, and equipment and this should be considered best practice. The study indicates that most attention should be paid to fomite sites with potential for direct contact with mucosal surfaces, or those protected from the elements but with potential for aerosol production, for example during cleaning. The greater frequency of detection of chlamydial DNA by qPCR than viable organism indicates the need to use viability assays in these studies, but further investment is needed to optimise these methods and improve sensitivity for the complex environmental samples encountered in wildlife rehabilitation facilities.

## ACKNOWLEDGEMENTS

The authors would like to thank the Koala Conservation Australia Ltd and Port Macquarie Koala Hospital staff and volunteers for their contributions and support during this project.

### Funding

This study was funded by the Australian Department of Agriculture, Water and the Environment: Regional Bushfire Recovery for Multiregional Species and Strategic Projects program. Andrea Casteriano is supported by the Wildlife Information, Rescue and Education Service (WIRES). The funders had no role in study design, data collection and analysis, decision to publish, or preparation of the manuscript.

### Grant Disclosures

The following grant information was disclosed by the authors:
Australian Department of Agriculture.
Water and the Environment: Regional Bushfire Recovery for Multiregional Species and Strategic Projects program.
Wildlife Information, Rescue and Education Service (WIRES).

### Competing Interests

The authors declare that they have no competing interests.

### Author Contributions

- Andrea Casteriano conceived and designed the experiments, performed the experiments, analyzed the data, prepared figures and/or tables, authored or reviewed drafts of the article, and approved the final draft.

- Astrid Robin Van Aggelen performed the experiments, authored or reviewed drafts of the article, and approved the final draft.
- Shali Fischer performed the experiments, authored or reviewed drafts of the article, and approved the final draft.
- Cheyne Flanagan performed the experiments, authored or reviewed drafts of the article, and approved the final draft.
- Caroline Marschner analyzed the data, prepared figures and/or tables, authored or reviewed drafts of the article, and approved the final draft.
- Hannah Nugent performed the experiments, analyzed the data, authored or reviewed drafts of the article, and approved the final draft.
- Wilhelmina Huston conceived and designed the experiments, authored or reviewed drafts of the article, and approved the final draft.
- Damien P. Higgins conceived and designed the experiments, analyzed the data, prepared figures and/or tables, authored or reviewed drafts of the article, and approved the final draft.

## Human Ethics

The following information was supplied relating to ethical approvals (*i.e.*, approving body and any reference numbers):

This study was approved by the Human Research Ethics Committee, University of Sydney (Approval No. 2022/007).

## Data Availability

Raw data are available in a Supplemental File.

## Supplemental Information

Supplemental information for this article can be found online at http://dx.doi.org/10.7717/peerj.15842#supplemental-information.

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
