# Peer review of "Evaluation of a biosecurity survey approach for contamination by *Chlamydia pecorum* in koala rehabilitation, field capture, and captive settings"

_PeerJ, doi:10.7717/peerj.15842_

## Round 0.1 · original submission · Major Revisions

Dear Dr. Casteriano and colleagues:

Thanks for submitting your manuscript to PeerJ. I have now received two independent reviews of your work, and as you will see, the reviewers raised some concerns about the research. Despite this, these reviewers are optimistic about your work and the potential impact it will have on research studying Chlamydia pecorum infections of koala. Thus, I encourage you to revise your manuscript, accordingly, taking into account all of the concerns raised by both reviewers.

While the concerns of the reviewers are relatively minor, this is a major revision to ensure that the original reviewers have a chance to evaluate your responses to their concerns (if deemed necessary). There are helpful suggestions, which I am sure will greatly improve your manuscript once addressed.

Please note that reviewer 2 has included a marked-up version of your manuscript.

I look forward to seeing your revision, and thanks again for submitting your work to PeerJ.

Good luck with your revision,

-joe

Reviewer 1 ·

Basic reporting

Line 54: Does the statement “chlamydiosis being the second most common reason for hospital admission after trauma” only refer to NSW koala admissions to wildlife hospitals? Refer to Gonzalez-Astudillo et al. 2017.
Line 55: Provide reference for infertility and population decline statement.
Line 55: Sentence beginning with “Chlamydial strain diversity….” Requires context to the relevance of your study.
Line 88: The UTM Transport Media swab is a COPAN product distributed by Interpath. Please correct swab reference.
Line 110: As per above comment for Line 88, SensiFAST. Probe No-ROX is manufactured by Bioline, not Meridian Bioscience. Please correct reference.
Line 124: Reference for established chlamydial culture system.

Experimental design

Line 117: Is the LOD the same for all 3 gene targets?
Line 126: Why was the infection stopped at 44 hours? Please describe if this timepoint is influenced by the development cycle of C. pecorum.

Validity of the findings

No comment

Additional comments

Very worthwhile and informative pilot study. As the manuscript suggests, further investigation of disinfection practices is warranted.

Reviewer 2 ·

Basic reporting

In this paper Casteriano et al describe and evaluate contamination by Chlamydia pecorum in koala rehabilitation, field capture, and captive settings (fomites/sites/equipment) by screening for both DNA and C pecorum viable cells (isolates). The paper is very interesting and important contribution to the field as not many are evaluating this aspect.
The article is clear and technically correct, contains aim/objective, relevant results (Tables) and discussion pertaining to those results.

I have some comments for clarity.

Experimental design

Its well designed study with methods described with sufficient detail & information to replicate.
Some minor comments for explanation of methodology used.

Validity of the findings

All underlying data have been provided; they are robust, statistically sound, & controlled. Discussion is linked to findings and objectives. Some minor comments for clarity.

Additional comments

Please see PDF with some specific in text comments.

Annotated reviews are not available for download in order to protect the identity of reviewers who chose to remain anonymous.

---

## Round 0.2 · accepted · Accept

Dear Dr. Casteriano and colleagues:

Thanks for revising your manuscript based on the concerns raised by the reviewers. I now believe that your manuscript is suitable for publication. Congratulations! I look forward to seeing this work in print, and I anticipate it being an important resource for groups studying Chlamydia pecorum infections of koala. Thanks again for choosing PeerJ to publish such important work.

Best,

-joe

Reviewer 1 ·

Basic reporting

No comment

Experimental design

No comment

Validity of the findings

No comment

Reviewer 2 ·

Basic reporting

No comment

Experimental design

no comment

Validity of the findings

no comment

Additional comments

I thank the authors for their revision. Manuscript is fantastic.
This is really interesting and important koala practice (biosecurity) study.
Congratulations on this study.